# NeurOLight: A Physics-Agnostic Neural Operator Enabling Parametric Photonic Device Simulation

**Jiaqi Gu[1], Zhengqi Gao[2], Chenghao Feng[1], Hanqing Zhu[1],**
**Ray T. Chen[1], Duane S. Boning[2], David Z. Pan[1]**
[1]The University of Texas at Austin, [2]Massachusetts Institute of Technology
*{jqgu, fengchenghao1996, hqzhu}@utexas.edu, zhengqi@mit.edu, boning@mtl.mit.edu,*
*{chen, dpan}@ece.utexas.edu*

## Abstract

Optical computing is an emerging technology for next-generation efficient artificial intelligence (AI) due to its ultra-high speed and efficiency. Electromagnetic field simulation is critical to the design, optimization, and validation of photonic devices and circuits. However, costly numerical simulation significantly hinders the scalability and turn-around time in the photonic circuit design loop. Recently, physics-informed neural networks have been proposed to predict the optical field solution of a single instance of a partial differential equation (PDE) with pre-defined parameters. Their complicated PDE formulation and lack of efficient parametrization mechanisms limit their flexibility and generalization in practical simulation scenarios. In this work, *for the first time*, a physics-agnostic neural operator-based framework, dubbed `NeurOLight`, is proposed to learn a family of frequency-domain Maxwell PDEs for ultra-fast parametric photonic device simulation. We balance the efficiency and generalization of `NeurOLight` via several novel techniques. Specifically, we discretize different devices into a unified domain, represent parametric PDEs with a compact wave prior, and encode the incident light via masked source modeling. We design our model with parameter-efficient cross-shaped `NeurOLight` blocks and adopt superposition-based augmentation for data-efficient learning. With these synergistic approaches, `NeurOLight` generalizes to a large space of unseen simulation settings, demonstrates *2-orders-of-magnitude* faster simulation speed than numerical solvers, and outperforms prior neural network models by ∼54% lower prediction error with ∼44% fewer parameters. Our code is available at link.

## 1 Introduction

With recent advances in integrated photonics technology, optical deep learning represents a new paradigm in next-generation efficient artificial intelligence (AI) [28, 27, 3]. An increasing number of co-design efforts have been made to enable synergistic *light-AI interaction*. We see extensive developments for photonic AI with rapidly evolving optical neural network (ONN) hardware accelerator designs [28, 8, 39, 5, 23, 29, 6] and various circuit-algorithm co-optimization methodologies [8, 9, 7, 30, 10, 14]. However, *applying AI for optics* is much less explored. Complementary to prior AI-assisted architectural exploration [19, 11], a natural question is *whether AI can assist in the lower-level device simulation* that requires a deep understanding of the physical nature of optics.

AI-assisted photonic device simulation is a critical step to closing the synergistic loop of *light-AI interaction*. Besides using standard devices that already have a compact transfer matrix [28, 8, 32], modern optical AI shows a trend to exploit customized photonic structures for scalable optical computing [7, 30, 40, 36]. Unfortunately, because customized devices do not have analytical transfer

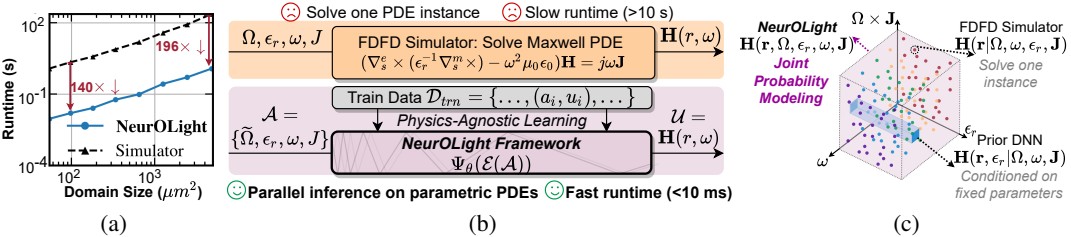

Figure 1: (a) Compare FDFD simulation and our `NeurOLight` framework. (b) `NeurOLight` (1 Quadro RTX 6000 GPU) runs 140×-200× faster than FDFD simulator (8-core i7-9700 CPUs) across different domain sizes (50 $nm$ grid step). (c) Different methods cover different solution space.

functions, understanding their behavior heavily relies on numerical simulators [16] to solve Maxwell partial differential equations (PDEs) to obtain the optical field distribution. Even solving a single 2-dimensional (2-D) simulation instance on a $5 \times 20~\mu m^2$ region can cost nearly 4 $s$ on 8 CPUs, as shown in Figure 1(a), which significantly hinders scalable circuit-level simulation and optimization. Hence, our target is to propose a surrogate model that learns the *light propagation principle* and efficiently approximates the field solutions to new simulation instances while running *orders-of-magnitude faster* than the numerical solver, as shown in Figure 1(b).

Most prior work still uses conventional NNs to predict several key properties based on a few design variables [31, 35], which is an ad-hoc function approximator without learning the light propagation property. Several works attempt to leverage physics-informed NNs (PINNs) [33, 2, 22] with physics-augmented residual loss to predict electromagnetic field solutions. However, previous methods have three major limitations. First, as illustrated in Figure 1(c), they only model the field distribution conditioned on pre-defined solving domains, input sources, and frequency. In other words, their models only learn the solution of a certain PDE instance with fixed parameters. Second, they all belong to the category of PINNs [25] that require an explicit description of the PDE as well as strict initial/boundary conditions. Constructing complicated Maxwell equation-based residual loss is no easier than re-implementing a numerical solver [24, 20, 21]. Third, their CNN-based models show inadequate modeling capacity with too small receptive fields to learn important light propagation, scattering, and interference effects.

To learn *a family of parametric Maxwell PDEs* that models the *joint probability* of different design variables as shown in Figure 1(c), we propose a physics-agnostic light field prediction framework `NeurOLight` that consists of a joint PDE encoder and an efficient cross-shaped neural operator backbone. The main contributions of the work are as follows:

- *For the first time*, an AI-based photonic device simulation framework is proposed to *learn a family of parametric* Maxwell PDEs for ultra-fast optical field prediction.

- We propose a novel joint PDE encoder for compact PDE representation and an efficient cross-shaped Fourier neural operator backbone for end-to-end optical field prediction, over **2-order-of-magnitude faster** than numerical simulators.

- We propose a superposition-based mixup technique that dynamically boosts the data efficiency and generalization during the training of `NeurOLight`.

- On different photonic device simulation benchmarks, our `NeurOLight` achieves the best prediction fidelity, generalizability, and transferability, outperforming UNet and state-of-the-art (SoTA) Fourier neural operators by an average of 53.8% lower prediction error with 44.2% fewer parameters.

- To our best knowledge, this is *the first* AI-based framework that can learn the terahertz light propagation inside photonic devices that generalizes to different domains, permittivities, input sources, and frequencies. We open-source our `NeurOLight` framework at link.

## 2   Related Work

**Optical field simulation with machine learning**.   Finite difference frequency domain (FDFD) simulation is a widely adopted method to analyze silicon-photonic devices. Numerical simulators

are used to solve frequency-domain Maxwell PDEs to obtain electromagnetic field distributions of an optical component with terahertz incident light sources. To accelerate this time-consuming process, NNs have been utilized as surrogate models for fast optical simulation approximation. A multi-layer perceptron was used to map the design variables to a scalar performance metric of a power splitter [31]. NNs were also utilized to predict intermediate values to accelerate the convergence of the numerical solver [35]. MaxwellNet [22] was proposed to train a UNet with physics-informed loss to predict the scattered field based on the material permittivity of the free-space lens. WaveYNet [2] also adopted a 2-D UNet as the model trained with both data-driven supervision loss and Maxwell-equation-based physics-augmented loss to predict the optical field of silicon meta-lens. However, prior NN-based methods require explicit physics knowledge of the Maxwell PDEs and only learn a small field solution space conditioned on fixed parameters.

**Learning PDEs via neural operators**. Recently, neural operators have been proposed as new NN models that learn a family of parametric PDEs in the infinite-dimensional function space in a mesh-free and purely data-driven fashion. The Fourier neural operator (FNO) [21] approximates the nonlinear mapping from PDE observations to solutions through Fourier-domain kernel integral operations, achieving record-breaking performance and efficiency on a wide range of challenging applications. Several variants have been proposed to improve the performance and efficiency of the original FNO models, e.g., Factorized FNO [34], U-FNO [37], and multiwavelet-based neural operator [12].

# 3 Neural operator-based optical field simulation framework `NeurOLight`

## 3.1 Understanding optical field simulation for photonic devices

Waveguides can confine the incident laser beam and allow the optical signals to propagate and interfere with each other. Various optical components, e.g., couplers, shifters, and multi-mode interference (MMI) devices [4], can create phase shifts, magnitude modulation, and interference, especially useful for optical communication and neuromorphic computing. Analyzing how light wave propagates through those components are critical to device optimization and photonic integrated circuit design. Given a linear isotropic optical component, we will shine time-harmonic continuous-wave light beam on its input ports and analyze the steady-state electromagnetic field distributions $\mathbf{E} = \hat{\mathbf{x}}\mathbf{E}_x + \hat{\mathbf{y}}\mathbf{E}_y + \hat{\mathbf{z}}\mathbf{E}_z$ and $\mathbf{H} = \hat{\mathbf{x}}\mathbf{H}_x + \hat{\mathbf{y}}\mathbf{H}_y + \hat{\mathbf{z}}\mathbf{H}_z$ in it, each of which includes horizontal ($x$), vertical ($y$), and longitudinal ($z$) components. We can solve the steady-state optical field $\mathbf{E}(\boldsymbol{r})$ and $\mathbf{H}(\boldsymbol{r})$ from the frequency-domain *curl-of-curl* Maxwell PDE under absorptive boundary conditions [16] (details in Appendix A1),

$$\big((\mu_0^{-1}\nabla \times \nabla \times) - \omega^2\epsilon_0\epsilon_r(\boldsymbol{r})\big)\mathbf{E}(\boldsymbol{r}) = j\omega\mathbf{J}_e(\boldsymbol{r}), \ \big(\nabla \times (\epsilon_r^{-1}(\boldsymbol{r})\nabla\times) - \omega^2\mu_0\epsilon_0\big)\mathbf{H}(\boldsymbol{r}) = j\omega\mathbf{J}_m(\boldsymbol{r}) \quad (1)$$

where $\nabla\times$ is the curl operator of a vector function, $\mu_0$ is the vacuum magnetic permeability, $\epsilon_0$ and $\epsilon_r$ are the vacuum and relative electric permittivity, and $\mathbf{J}_m$ and $\mathbf{J}_e$ are the magnetic and electric current sources. FDFD simulation discretizes the continuous domain into an $M \times N$ mesh grid and solves the above linear equation $\mathbf{AX} = \boldsymbol{b}$ to obtain the fields. Detailed forms of $\mathbf{A}$ and $\boldsymbol{b}$ can be found in [16]. Solving for these optical fields exactly with a sparse matrix $\mathbf{A} \in \mathbb{C}^{MN \times MN}$ is prohibitively expensive and not scalable to large photonic structures. A fast surrogate model that predicts optical fields with high fidelity is of tremendous interest.

## 3.2 The proposed `NeurOLight` framework

As shown in Figure 2, our `NeurOLight` framework models the optical field simulation problem as an infinite-dimensional-space mapping from Maxwell PDE observations $\mathcal{A} \in \mathbb{C}^{\Omega \times d_a}$ to the optical field solution $\mathcal{U} \in \mathbb{C}^{\Omega \times d_u}$. Here, $\Omega$ is the continuous 2-D physical solving domain, $\Omega = (l_x, l_z)$, typically in units of micrometers ($\mu m$), where the photonic device-of-interest can be tightly located. $\mathcal{A}$ and $\mathcal{U}$ take values with $d_a$ and $d_u$ dimensions, respectively. To learn the ground truth nonlinear mapping $\Psi^* : \mathcal{A} \to \mathcal{U}$, we construct `NeurOLight` with a PDE encoder $\mathcal{E}$ that produces compact PDE representations, followed by an efficient neural operator-based approximator $\Psi_\theta$ that minimizes the empirical error on discrete PDE observable samples $a \sim \mathcal{A}$,

$$\theta^* = \min_\theta \mathbb{E}_{a\sim\mathcal{A}}\big[\mathcal{L}\big(\Psi_\theta(\mathcal{E}(a)), \Psi^*(a)\big)\big]. \quad (2)$$

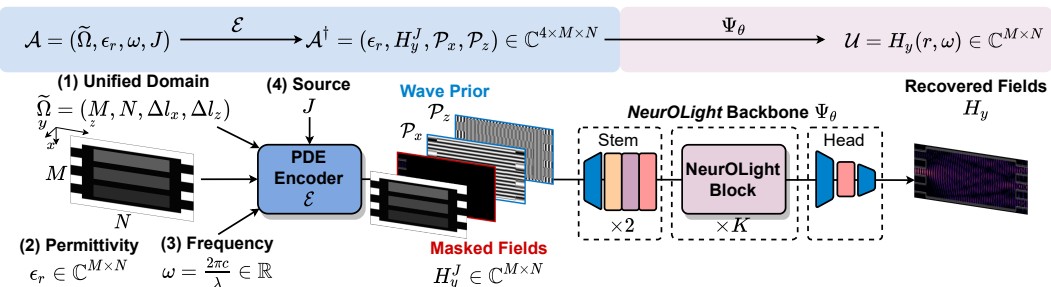

Figure 2: `NeurOLight` framework for optical field simulation. Real part is plotted for complex fields.

### 3.2.1 Scale-adaptive domain discretization: $\Omega \rightarrow \widetilde{\Omega}$

To generalize to PDEs in *different physical domains* and support batched parallel inference, we adopt an $M \times N$ discrete unified domain $\widetilde{\Omega} = (M, N, \Delta l_x, \Delta l_z)$ with an *adaptive mesh granularity*, i.e., with grid steps $\Delta l_x = l_x/M$ and $\Delta l_z = l_z/N$. As shown in Figure. 3, multiple photonic devices with different physical dimensions are normalized to the same $\widetilde{\Omega}$. Their original physical dimensions can be elegantly encoded into the re-calculated mesh granularities. This unified discrete domain gives `NeurOLight` the flexibility to handle parallel inference on different physical domain dimensions, unlike prior work [22, 2] that requires time-consuming model retraining once the physical domain changes.

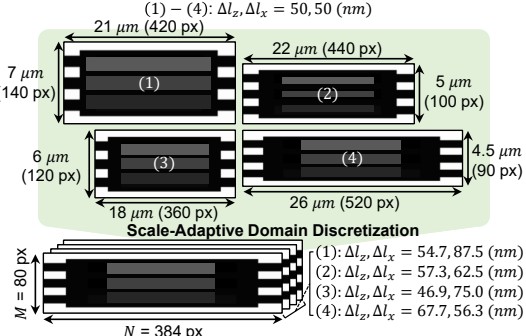

Figure 3: Scale-adaptive domain discretization enables generalization to different solving domain dimensions and efficient batched processing.

### 3.2.2 Joint PDE representation: $\mathcal{A} \rightarrow \mathcal{A}^\dagger$

After we define a unified solving domain, we need to construct effective PDE representations that describe the raw observations $\mathcal{A} = (\widetilde{\Omega}, \boldsymbol{\epsilon}_r, \omega, \mathbf{J})$. The relative permittivity distribution can be simply represented by $\boldsymbol{\epsilon}_r \in \mathbb{C}^{M \times N}$. However, how to compactly encode other parameters, i.e., $(\widetilde{\Omega}, \omega, \mathbf{J})$, remains a non-trivial challenge. Let us first consider what makes a good representation. First, it needs to be compatible with the model input, i.e., it can be fused with the 2-D image representation in a *compact* way. Second, it is preferred to reveal the physical essence of the parameters and inject useful *prior knowledge* that helps model generalization. Based on the above considerations, we propose a PDE encoder $\mathcal{E} : \mathcal{A} \rightarrow \mathcal{A}^\dagger$ that converts the raw observations to a joint PDE representation $\mathcal{A}^\dagger = (\boldsymbol{\epsilon}_r, \mathbf{H}_y^J, \mathcal{P}_x, \mathcal{P}_z)$.

**Encoding $(\widetilde{\Omega}, \boldsymbol{\epsilon}_r, \omega)$ via wave prior.** The intuition behind the wave prior design is that the vacuum angular frequency $\omega = \frac{2\pi c}{\lambda}$ and electric permittivity $\boldsymbol{\epsilon}_r$ together decide the physical light wavelength inside the material, i.e., $\lambda' = \lambda/\sqrt{\epsilon_r}$. The mesh granularity determines how many pixels can depict a wave period along both directions, i.e., $(\lambda'/\Delta l_x, \lambda'/\Delta l_z)$. Therefore, as

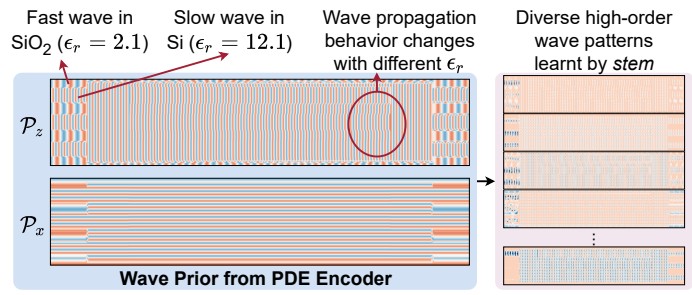

Figure 4: Wave prior as joint PDE representations.

shown in Figure 4, we construct artificial wave patterns, named *wave prior*, as $\mathcal{P}_z = e^{j \frac{2\pi \sqrt{\epsilon_r}}{\lambda} \mathbf{1} z^T \Delta l_z}$ and $\mathcal{P}_x = e^{j \frac{2\pi \sqrt{\epsilon_r}}{\lambda} x \mathbf{1}^T \Delta l_x}$, where $x = (0, 1, \cdots, M - 1)$ and $z = (0, 1, \cdots, N - 1)$. The wave

prior jointly translates the $(\widetilde{\Omega}, \epsilon_r, \omega)$ pair to a unified representation with strong prior knowledge, significantly reducing the learning difficulty on overly-abstract raw observations. We note that the `NeurOLight` stem learns complex combinations of the wave prior and generates diverse high-order wave patterns for later feature transformation.

**Masked image modeling-inspired light source (J) encoding.** In the optical simulation, *light source* **J** will be injected by shining light on the input waveguides of the photonic devices as stimuli to the system. **J** will have a vacuum angular frequency $\omega$ and a polarization mode. For example, in the transverse magnetic (TM) mode, we have $\mathbf{H}_x = \mathbf{H}_z = 0$. Thus we focus on the prediction of $\mathbf{H}_y$. However, as shown in Figure 5, **J** is a combination of multiple length-$w$ 1-D vectors, where $w$ is the input port width, placed at the input waveguides, which is hard to be encoded in the image prediction flow. Therefore, we borrow the idea of *masked image modeling* [1] to light source encoding. In the source representation $\mathbf{H}^J$, we only maintain the fields in the input waveguides before entering the key region of the device, which is easy to obtain and irrelevant to the structure it enters into, and mask out all the fields after. In this way, the field prediction task translates to a *masked field restoration* task conditioned on the input light source as a *hint*.

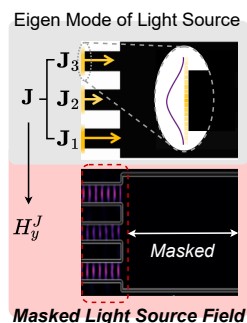

Figure 5: Masked light source modeling.

### 3.3 Efficient `NeurOLight` model architecture: $\Psi_\theta$

**Convolutional stem**. The proposed `NeurOLight` architecture starts with a convolutional stem $\mathcal{S}$ : $a^\dagger(\boldsymbol{r}) \to v_0(\boldsymbol{r}), \forall \boldsymbol{r} \in \Omega$ that encodes each complex-valued observation sample $a^\dagger(\boldsymbol{r}) \in \mathbb{C}^{4 \times M \times N}$ to a real-valued representation $v_0(\boldsymbol{r}) \in \mathbb{R}^{C \times M \times N}$. Lightweight blueprint convolutions [13] are used to perform local wave pattern transformation with a low hardware cost.

**Cross-shaped `NeurOLight` block.** In the projected $C$-dimensional space, we place $K$ cascaded `NeurOLight` blocks to gradually restore the complex light field in the frequency domain as $v_0(\boldsymbol{r}) \to v_1(\boldsymbol{r}) \to \cdots \to v_K(\boldsymbol{r})$. Each `NeurOLight` block is formulated as

$$v_{k+1}(\boldsymbol{r}) := \text{FFN}\big((\mathcal{K}v_k)(\boldsymbol{r})\big) + v_k, \forall \boldsymbol{r} \in \Omega; \quad (\mathcal{K}v_k)(\boldsymbol{r}) = \int_\Omega \kappa(\boldsymbol{r}_1, \boldsymbol{r}_2)v_k(\boldsymbol{r}_2)\mathrm{d}v_k(\boldsymbol{r}_2), \forall \boldsymbol{r}_1 \in \Omega, \quad (3)$$

where $\mathcal{K}$ is a learnable kernel integral transform, and $\text{FFN}(\cdot)$ is a feedforward network. When the kernel satisfies $\kappa(\boldsymbol{r}_1, \boldsymbol{r}_2) = \kappa(\boldsymbol{r}_1 - \boldsymbol{r}_2)$, the above integral kernel operator is equivalent to a spatial-domain 2-D convolution, which can be efficiently computed by using Fourier transform $\mathcal{F}(\cdot)$ [21].

A clear downside of the original 2-D FNO is the huge parameter cost, i.e., $\mathcal{F}(\kappa)(\boldsymbol{r}) \in \mathbb{C}^{k_v \times k_h \times C \times C}$, and the resultant severe overfitting issues. To improve the model *efficiency and generalization* simultaneously, we introduce a *cross-shaped Fourier neural operator*, shown in Figure 6. The input feature is first bi-partitioned along the channel dimension into two chunks $v_k(\boldsymbol{r}) = [v_k^h(\boldsymbol{r}); v_k^v(\boldsymbol{r})]$, representing horizontal and vertical patterns, and 1-D FNO is applied to both directions,

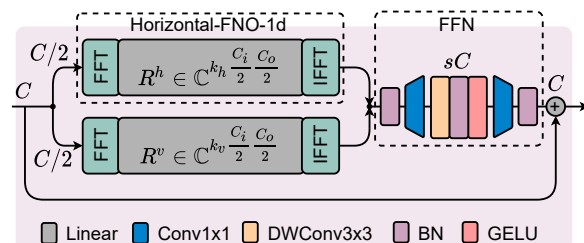

Figure 6: `NeurOLight` backbone model design.

$$(\mathcal{K}^h v_k^h)(\boldsymbol{r}) = \mathcal{F}_z^{-1}\big(\mathcal{F}_z(\kappa^h) \cdot \mathcal{F}_z(v_k^h)\big)(\boldsymbol{r}) = \mathcal{F}_z^{-1}\big(R^h(z) \cdot \mathcal{F}_z(v_k^h(\boldsymbol{r}))\big), \forall z \in \Omega_z, \forall r \in \Omega,$$
$$(\mathcal{K}^v v_k^v)(\boldsymbol{r}) = \mathcal{F}_x^{-1}\big(\mathcal{F}_x(\kappa^v) \cdot \mathcal{F}_x(v_k^v)\big)(\boldsymbol{r}) = \mathcal{F}_x^{-1}\big(R^v(x) \cdot \mathcal{F}_x(v_k^v(\boldsymbol{r}))\big), \forall x \in \Omega_x, \forall r \in \Omega, \quad (4)$$
$$(\mathcal{K}v_k)(\boldsymbol{r}) = [(\mathcal{K}^h v_k^h)(\boldsymbol{r}); (\mathcal{K}^v v_k^v)(\boldsymbol{r})].$$

We parametrize the Fourier kernels as lightweight complex-valued tensors $R^h(z) \in \mathbb{C}^{k_z \times \frac{C}{2} \times \frac{C}{2}}$ and $R^v(x) \in \mathbb{C}^{k_x \times \frac{C}{2} \times \frac{C}{2}}$. These orthogonal 1-D kernel operations intrinsically perform spatial and channel-wise feature aggregation in an interleaved way that are able to provide global receptive fields to achieve long-distance modeling. Compared with $k_v k_h C^2$ parameters in the 2-D FNO, our cross-shaped `NeurOLight` block only has $\frac{(k_v + k_h + 8s)C^2}{4}$ parameters.

To increase nonlinearity and enhance local information interaction, we append an FFN block after the cross-shaped FNO. Inspired by the MixFFN designs in SoTA vision transformers [38], our FFN expands the channels by $s$ times, performs local information aggregation via 3×3 depth-wise convolution (DWConv), activates using the GELU function, and projects it back to $C$ channels.

**Projection head**. At the end, two point-wise convolutional layers are used to project $v_K(\boldsymbol{r})$ to the light field space $u(\boldsymbol{r}) = \mathcal{Q}(v_K(\boldsymbol{r}))$. Dropout layer is inserted to mitigate overfitting issues.

**Loss function**. Even with normalized light source power, optical fields tend to have distinct statistics. To balance the optimization effort among different fields, we adopt the normalized mean absolute error (N-MAE) as the objective $\mathcal{L}\big(\Psi_\theta(\mathcal{E}(a)), \Psi^*(a)\big) = (\|\Psi_\theta(\mathcal{E}(a)) - \Psi^*(a)\|_1)/\|\Psi^*(a)\|_1$.

### 3.4 Toward better data efficiency and generalization via superposition-based mixup

The PDE observations $\mathcal{A}$ can cover a huge design space. Hence, the data efficiency and generalization of pure data-driven models raise a concern. Simply drawing large numbers of random training examples with all possible light sources has an intractable data acquisition cost. Standard augmentation techniques are effective in improving data efficiency and generalization on tasks with *natural images*; however, their direct application is *not compatible with PDE simulation*. Since $\Psi^*(a_i)$ is a highly nonlinear function of $a_i$ and closely related to the boundary conditions, simultaneously augmenting $\boldsymbol{\epsilon}_r$, $\Omega$, $\omega$, and $\mathbf{H}$, e.g., via cropping, distortion, or resizing, leads to *invalid* field solutions.

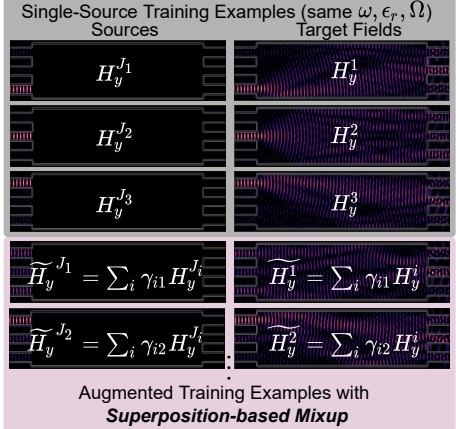

Interestingly, we notice that the photonic system satisfies the *superposition* property w.r.t. the light source,

$$\widetilde{\mathbf{H}} = \Psi^*(\widetilde{\mathbf{H}}^J) = \Psi^*(\sum_{i=1}^{|\mathbf{J}|} \gamma_i \mathbf{H}^{J_i}) = \sum_{i=1}^{|\mathbf{J}|} \gamma_i \Psi^*(\mathbf{H}^{J_i}). \quad (5)$$

Figure 7: Data augmentation with superposition-based mixup. Only real part is plotted for each field.

Based on this, we only involve *single-input simulation data* in the training set and dynamically mix multiple ($|\mathbf{J}|$) input sources via *Superposition-based Mixup*, as shown in Figure 7,

$$\begin{pmatrix} \widetilde{\mathbf{H}}^{J_1} & \cdots & \widetilde{\mathbf{H}}^{J_{|\mathbf{J}|}} \\ \widetilde{\mathbf{H}}^1 & \cdots & \widetilde{\mathbf{H}}^{|\mathbf{J}|} \end{pmatrix}^T = \mathbf{\Gamma} \begin{pmatrix} \mathbf{H}^{J_1} & \cdots & \mathbf{H}^{J_{|\mathbf{J}|}} \\ \mathbf{H}^1 & \cdots & \mathbf{H}^{|\mathbf{J}|} \end{pmatrix}^T, \quad \mathbf{\Gamma} \in \mathbb{C}^{|\mathbf{J}|\times|\mathbf{J}|}, \|\mathbf{\Gamma}_j\|_2 = 1, \phi(\gamma_{j1}) = 0, \forall j \in [|\mathbf{J}|]. \quad (6)$$

At each iteration, we randomly generate the mixup coefficient matrix $\mathbf{\Gamma}$ and make it have a unit row-wise $\ell_2$-norm for light field power normalization. The global phase of the complex-valued optical field is normalized by forcing the first input port always to have a phase that equals 0. In this way, `NeurOLight` learns how multiple incident light sources interfere with one another. Once `NeurOLight` can generalize to arbitrary light sources, multi-source simulation only needs *an efficient one-shot inference with superposed source fields* instead of explicitly accumulating $|\mathbf{J}|$ single-source inference results.

## 4 Results

### 4.1 Experiment setup

**Datasets.** We focus on widely applied multi-mode interference (MMI) photonic devices. We select MMIs with rectangular tunable control pads (*Tunable MMI*). The permittivity of the tuning region can be programmed by external signals, so this family of devices exemplifies photonic structures with reconfigurable transmissions [18]. We also evaluate MMIs with rectangular etched cavities (*Etched MMI*) [31], which exemplifies another popular category of passive sub-wavelength photonic devices with fixed yet highly discrete permittivity distributions. We use an open-source CPU-based 2-D FDFD simulator angler [16] to simulate optical fields for randomly generated MMIs as our dataset. Details on dataset generation is in Appendix A2.

## 4.2 Main results

In Table 1, we compare four models: (1) UNet [22, 2], (2) a 5-layer FNO-2d [21], (3) a 12-layer factorized FNO (F-FNO) [34], and (4) our `NeurOLight`. Detailed training settings and model architectures can be found in Appendix A3 and Appendix A4, respectively. On these benchmarks, `NeurOLight` outperforms UNet and prior SoTA FNO variants by **53.8%** lower test error with **44.2%** fewer parameters on average.

**Results on tunable MMI**. On tunable MMI, `NeurOLight` achieves the best prediction error with only half the parameter cost. Figure 8 visualizes the field prediction for one test MMI. UNet is significantly limited by its small receptive field and lack of long-distance modeling capability, thus failing to predict the full field even with the hint of wave prior. As representative neural operators, FNO-2d and factorized FNO

Table 1: Comparison of parameter count, train error, and test error on two benchmarks among four different models.

| Benchmarks | Model | #Params (M) ↓ | Train Err ↓ | Test Err ↓ |
|---|---|---|---|---|
| Tunable MMI | UNet [22, 2] | 3.47 | 0.776 | 0.801 |
| | FNO-2d [21] | 3.29 | 0.231 | 0.244 |
| | F-FNO [34] | 3.16 | 0.272 | 0.292 |
| | `NeurOLight` | **1.58** | **0.145** | **0.122** |
| Etched MMI | UNet [22, 2] | 3.47 | 0.779 | 0.792 |
| | FNO-2d [21] | 3.29 | 0.601 | 0.648 |
| | F-FNO [34] | 3.16 | 0.411 | 0.525 |
| | `NeurOLight` | **2.11** | **0.376** | **0.387** |
| Average Improvement | | **-44.2%** | **-49.1%** | **-53.8%** |

(F-FNO) manifest the superior advantages of the Fourier-domain kernel integral operations, showing considerably lower prediction errors than their CNN counterparts. However, given the parameter budget (∼3 M), the 5-layer FNO-2d only has 10 modes in the $x$-direction and 32-modes in the $z$-direction, which may not be enough to extract high-frequency waves. The 12-layer F-FNO adopts factorized 1-D Fourier kernel to save parameters; however, its modeling capability is limited by the lack of local feature extractors. Our `NeurOLight` blocks benefit from the global view of the cross-shaped 1-D kernel and local feature aggregation from convolutional FFN blocks. In the training curves in Figure 10(a), `NeurOLight` achieves the fastest convergence and best generalization among all models. We animate the *real-time* prediction process of `NeurOLight` in Appendix A5.

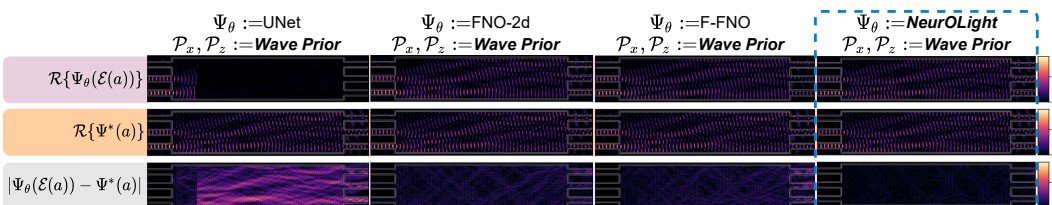

Figure 8: Visualization on one test tunable MMI. ($\Delta l_x = 83.1\ nm, \Delta l_z = 70.8\ nm, \lambda = 1.54\ \mu m$).

**Results on etched MMI**. Compared with tunable MMIs, predicting the field in etched MMIs, even with 2× more training examples, is a much harder task given the complicated scattering at the cavity-silicon interface and the considerably larger and highly discrete design space, shown in Figure 9. Hence, we increase the model capacity of `NeurOLight` by using 16 layers. Among all prediction models, `NeurOLight` achieves the best results with 42% lower error while still saving 36% parameters on average.

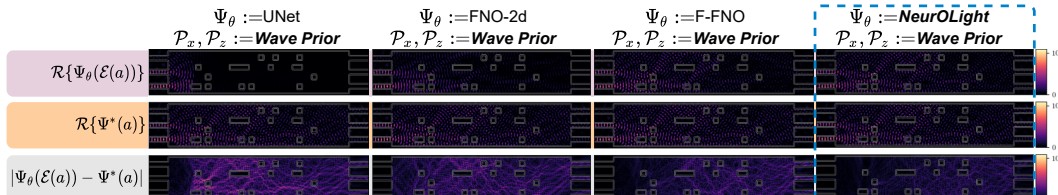

Figure 9: Visualization on one test etched MMI. ($\Delta l_x = 91.3\ nm, \Delta l_z = 89.1\ nm, \lambda = 1.55\ \mu m$).

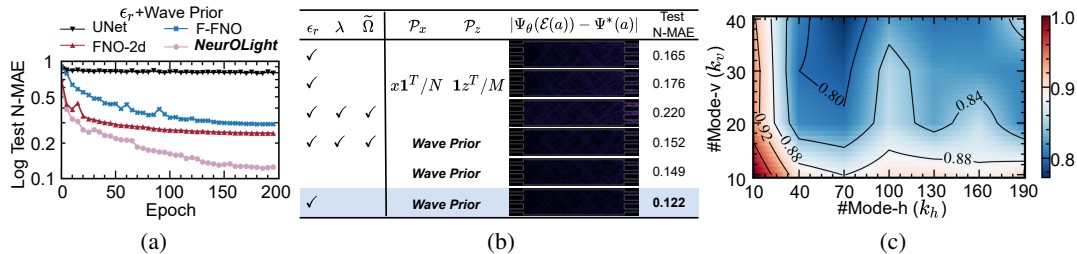

Figure 10: (a) Test N-MAE curves of four models. (b) Our PDE encoder achieves the lowest error. (c) Normalized test error contour of a 8-layer `NeurOLight` with different # of Fourier modes.

## 4.3 Ablation studies

**PDE encoding**. In Figure 10(b), we extensively compare different combinations of PDE encoding methods. The first two methods only model the distribution over $\epsilon_r$ conditioned on fixed wavelength and domain sizes like prior work [22, 2], which fail to generalize to examples in larger design space. With raw PDE parameters ($\epsilon_r, \lambda, \widetilde{\Omega}$), the model finds it difficult to learn a generalizable representation, thus showing large errors on the test dataset. The last three combinations validate that permittivity and our wave prior are compact and effective encodings in our joint PDE representation method, while extra raw wavelength and domain information are redundant and even harmful.

**Fourier modes**. As shown in Figure 10(c), we perform a fine-grained exploration in the Fourier mode space to find the most suitable configuration. Unlike the flow prediction tasks evaluated in FNO [21] that only require a few modes, using inadequate Fourier modes fails to learn the terahertz high-frequency optical fields in the photonic device simulation task. However, using all Fourier series is not necessary and makes the model prone to overfitting issues. $(k_h, k_v)$=(70, 40) is the best setting that balances expressiveness and efficiency in our `NeurOLight`.

**Cross-shaped `NeurOLight` block**. In Table 2, we *independently* change one technique in the full `NeurOLight` model to verify each individual contribution. Our proposed essential techniques *synergistically* boost the modeling capacity and generalization. Compared to the linear lifting in FNO-2d that only performs point-wise projection, our lightweight convolution stem can extract complex high-order wave patterns with negligible runtime

Table 2: Ablation on proposed techniques. Each entry changes one technique independently. Runtime is averaged over multiple runs on 1 NVIDIA Quadro RTX 6000 GPU.

| Variants | #Params (M)↓ | Train Err ↓ | Test Err ↓ | Runtime (ms) ↓ |
|---|---|---|---|---|
| **NeurOLight** | **1.58** | **0.145** | **0.122** | **12.1** |
| ConvStem → Lifting | 1.58 | 0.156 | 0.134 | 11.9 |
| Extra Parallel Conv Path | 1.64 | 0.149 | 0.129 | 14.5 |
| FFN → BN-GELU | 1.37 | 0.469 | 0.446 | 6.3 |
| Remove DWConv in FFN | 1.57 | 0.164 | 0.144 | 10.6 |
| Extra GELU After FNO | 1.58 | 0.164 | 0.148 | 12.4 |
| Remove DropPath | 1.58 | 0.131 | 0.136 | 12.1 |

overhead. Similar to U-FNO [37], we append an additional parallel convolution path alongside the cross-shaped FNO block; however, the extra 20% runtime penalty does not pay off. The proposed convolutional FFN significantly improves the nonlinearity and local feature extraction ability of `NeurOLight`. Changing it to a simple BatchNorm-GELU causes significant degradation. Different from the MLP-based FFN in F-FNO [34], our extra depthwise CONV is critical to local feature extraction and can reduce the test error by 16%. Note that an extra GELU after the FNO block will distort the feature in the low-dimensional space and have a negative impact on the performance [26]. Besides the dropout in the head, stochastic network depth [15] in the residual `NeurOLight` block is also effective in mitigating overfitting.

## 4.4 Discussion

**Superposition-based mixup.**

As shown in Table 3, without augmentation, `NeurOLight` only sees single-source training examples, thus failing to generalize when multiple sources are fused as a unified input source for *fast one-shot* prediction, named multi-source inference mode. A simple work-around would be to perform single-source prediction on $|\mathbf{J}|$ ports and superpose the resultant $|\mathbf{J}|$ fields, named single-source inference

mode. When training on a large enough training set, this method indeed works. However, it quickly deteriorates as the training set reduces with a $|\mathbf{J}|$ times higher runtime cost for a $|\mathbf{J}|$-port device. With our dynamic superposition-based mixup, `NeurOLight` works well both in single-source and multi-source inference modes with superior generalizability even with only 10% training data.

Table 3: Test N-MAE of an 8-layer `NeurOLight` with different number of training examples. Multi-source inference mode has similar performance as the single-source method but shows $3\times$ faster runtime on $3\times3$ MMIs.

| Train Augmentation | Inference Mode | #Train Examples (K) | | | | | Runtime (ms) |
|---|---|---|---|---|---|---|---|
| | | 1.4 | 4.1 | 6.9 | 9.7 | 12.4 | |
| None | Single-Source | 0.346 | 0.257 | 0.202 | 0.198 | 0.194 | 23.8 |
| | Multi-Source | 0.892 | 0.882 | 0.880 | 0.865 | 0.873 | 8.3 |
| **Superposition Mixup** | Single-Source | 0.229 | 0.205 | 0.204 | 0.200 | 0.199 | 23.8 |
| | **Multi-Source** | **0.230** | **0.208** | **0.206** | **0.202** | **0.202** | **8.3** |

**Spectrum analysis**. Spectroscopy is an important approach to understanding the broadband response of a photonic device. A traditional FDFD simulator has to sweep the spectrum with multiple simulations. In contrast, `NeurOLight` models the joint probability over wavelengths, and thus only needs to perform parallel inference with different $\omega = \frac{2\pi}{\lambda}$ values *at one shot*. Figure 11 demonstrates that, though `NeurOLight` is only trained with five selected wavelengths, it can generalize to unseen devices with unseen wavelengths. Sweeping in the standard C-band (1550 $nm$-1565 $nm$) with a 2 $nm$ granularity, `NeurOLight` can finish within 150 $ms$, achieving $450\times$ speedup over the FDFD simulator.

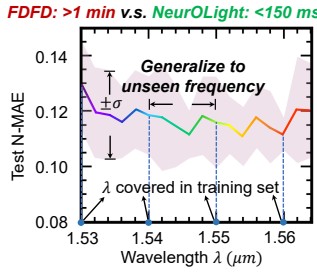

Figure 11: `NeurOLight` can generalize to unseen devices and wavelengths.

**Device adaptation**. We evaluate the transferability of `NeurOLight` via device adaptation. In Figure 12, we transfer `NeurOLight` trained on 3-port MMIs to larger MMIs with 4 and 5 ports. Directly predicting new devices shows unsatisfying test error out of distribution (OOD). We adapt the model with 20-epoch fast linear probing and 30-epoch finetuning [17] on 3.7 K 4-port MMI examples and 4.6 K 5-port MMI examples. The model quickly transfers to new photonic devices with good prediction fidelity.

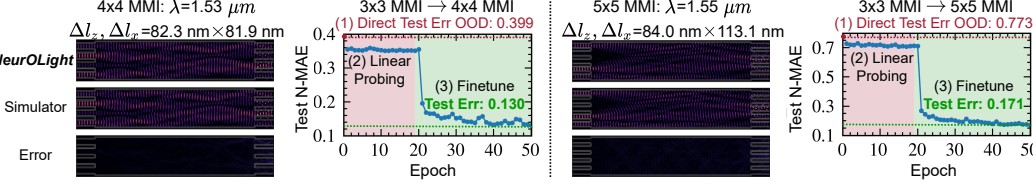

Figure 12: Device adaptation from 3-port to 4-/ 5-port MMI via linear probing and finetuning.

## 5 Conclusion

In this work, *for the first time*, a physics-agnostic neural operator, named `NeurOLight`, is proposed for ultra-fast parametric photonic device simulation. We propose a joint PDE encoder with wave prior and masked source modeling for compact PDE representation. Our lightweight cross-shaped `NeurOLight` backbone design achieves a superior balance between modeling capability and parameter efficiency. In addition, our novel superposition-based mixup technique significantly boosts the data efficiency and model generalizability. Experiments show that `NeurOLight` outperforms prior DNN models with 53.8% better prediction fidelity and 44.2% less parameter cost, serving as an over $100\times$ faster surrogate model to the numerical solvers in photonic device simulation. Currently, our model focuses on device-level simulation. As a future direction, we look forward to exploring the circuit-level simulation and utilizing our model to streamline the optimization loop for efficient AI-assisted optical circuit design automation.

**Acknowledgments** The authors acknowledge the Multidisciplinary University Research Initiative (MURI) program through the Air Force Office of Scientific Research (AFOSR), contract No. FA 9550-17-1-0071, monitored by Dr. Gernot S. Pomrenke.

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

## A1   Optical field simulation

Analyzing how light field propagate through those components are critical to device optimization and photonic integrated circuit design. Given a linear isotropic optical component, we will shine time-harmonic continuous-wave light on its input ports and analyze the steady-state electromagnetic field distributions $\mathbf{E} = \hat{\mathbf{x}}\mathbf{E}_x + \hat{\mathbf{y}}\mathbf{E}_y + \hat{\mathbf{z}}\mathbf{E}_z$ and $\mathbf{H} = \hat{\mathbf{x}}\mathbf{H}_x + \hat{\mathbf{y}}\mathbf{H}_y + \hat{\mathbf{z}}\mathbf{H}_z$ in it, each of which includes horizontal $(x)$, vertical $(y)$, and longitudinal $(z)$ components. The light field follows the Maxwell PDE under certain absorptive boundary conditions [16],

$$\nabla \times \mathbf{E}(\boldsymbol{r}, t) = \mu_0 \frac{\partial \mathbf{H}(\boldsymbol{r}, t)}{\partial t} + \mathbf{J}_e(\boldsymbol{r}, t), \ \nabla \times \mathbf{H}(\boldsymbol{r}, t) = -\epsilon_0 \epsilon_r(\boldsymbol{r}) \frac{\partial \mathbf{E}(\boldsymbol{r}, t)}{\partial t} + \mathbf{J}_e(\boldsymbol{r}, t), \tag{7}$$

where $\nabla \times$ is the curl operator of a vector function, $\mu_0$ is the vacuum magnetic permeability, $\epsilon_0$ and $\epsilon_r$ are the vacuum and relative electric permittivity, $\mathbf{J}_m$ and $\mathbf{J}_e$ are the magnetic and electric current sources. Since the input light is time-harmonic at a vacuum angular frequency $\omega$, the time-domain PDE can be transformed to the frequency domain for the steady state as follows,

$$\nabla \times \mathbf{E}(\boldsymbol{r}) = j\omega\mu_0\mathbf{H}(\boldsymbol{r}) + \mathbf{J}_m(\boldsymbol{r}), \ \nabla \times \mathbf{H}(\boldsymbol{r}) = -j\omega\epsilon_0\epsilon_r(\boldsymbol{r})\mathbf{E}(\boldsymbol{r}) + \mathbf{J}_e(\boldsymbol{r}). \tag{8}$$

A simple variable substitution gives us the *curl-of-curl* Maxwell PDE,

$$\left((\mu_0^{-1}\nabla \times \nabla\times) - \omega^2\epsilon_0\epsilon_r(\boldsymbol{r})\right)\mathbf{E}(\boldsymbol{r}) = j\omega\mathbf{J}_e(\boldsymbol{r}), \ \left(\nabla \times (\epsilon_r^{-1}(\boldsymbol{r})\nabla\times) - \omega^2\mu_0\epsilon_0\right)\mathbf{H}(\boldsymbol{r}) = j\omega\mathbf{J}_m(\boldsymbol{r}). \tag{9}$$

To restrict a unique solution without boundary reflection, complicated boundary conditions will be inserted [16]. An artificial material, i.e., coordinate-stretched perfectly matched layer (SC-PML), will be padded around the solving domain. Such PML materials have large imaginary parts in the permittivities to introduce strong energy absorption and changes the derivative operator to $\nabla = (\frac{1}{s_x(x)}\frac{\partial}{\partial x}, \frac{1}{s_y(y)}\frac{\partial}{\partial y}, \frac{1}{s_z(z)}\frac{\partial}{\partial z})$, where $s$ is a location-determined complex value. Solving the above PDEs will give the steady-state frequency-domain complex magnitude of the optical fields.

## A2   Dataset generation

We generate our customized MMI device simulation dataset using an open-source FDFD simulator angler [16]. The tunable MMI dataset has 5.5 K *single-source* training data, 614 validation data, and 1.5 K multi-source test data. The etched MMI dataset has 12.4 K *single-source* training data, 1.4 K validation data, and 1.5 K *multi-source* test data. We summarize how we generate random devices in Table A4. We randomly sample the physical dimension of the MMI, input/output waveguide width, the width of the perfectly matched layer (PML), device border width away from PML, controlling pad sizes, input light source frequencies, etched cavity sizes and ratio (determines the number of cavities in the MMIs), and permittivities in the controlling region.

Table A4: Summary of device design variable's sampling range, distribution, and unit.

| Variables | Value/Distribution | | Unit |
|---|---|---|---|
| | $|\mathbf{J}| \times |\mathbf{J}|$ Tunable MMI | $|\mathbf{J}| \times |\mathbf{J}|$ Etched MMI | |
| Length | $\mathcal{U}(20, 30)$ | $\mathcal{U}(20, 30)$ | $\mu m$ |
| Width | $\mathcal{U}(5.5, 7)$ | $\mathcal{U}(5.5, 7)$ | $\mu m$ |
| Port Length | 3 | 3 | $\mu m$ |
| Port Width | $\mathcal{U}(0.8, 1.1)$ | $\mathcal{U}(0.8, 1.1)$ | $\mu m$ |
| Border Width | 0.25 | 0.25 | $\mu m$ |
| PML Width | 1.5 | 1.5 | $\mu m$ |
| Pad Length | $\mathcal{U}(0.7, 0.9)\times$Length | $\mathcal{U}(0.7, 0.9)\times$Length | $\mu m$ |
| Pad Width | $\mathcal{U}(0.4, 0.65)\times$Width/$|\mathbf{J}|$ | $\mathcal{U}(0.4, 0.65)\times$Width/$|\mathbf{J}|$ | $\mu m$ |
| Wavelengths $\lambda$ | $\mathcal{U}(1.53, 1.565)$ | $\mathcal{U}(1.53, 1.565)$ | $\mu m$ |
| Cavity Ratio | - | $\mathcal{U}(0.05, 0.1)$ | - |
| Cavity Size | - | 0.027 Length $\times$ 0.114 Width | $\mu m^2$ |
| Relative Permittivity $\epsilon_r$ | $\mathcal{U}(11.9, 12.3)$ | $\{2.07, 12.11\}$ | - |

## A3   Training settings

We implement all models and training logic in PyTorch 1.10.2. All experiments are conducted on a machine with Intel Core i7-9700 CPUs and an NVIDIA Quadro RTX 6000 GPU. For training from

scratch, we set the number of epochs to 200 with an initial learning rate of 0.002, cosine learning rate decay, and a mini-batch size of 12. For the tunable MMI dataset, we split all 7,680 examples into 72% training data, 8% validation data, and 20% test data. For the etched MMI dataset, we split all 15,360 examples into 81% training data, 9% validation data, and 10% test data. For device adaptation, we first perform linear probing for 20 epochs with an initial learning rate of 0.002 and cosine learning rate decay; then we perform finetuning for 30 epochs with an initial learning rate of 0.0002 and a cosine learning rate decay. We apply stochastic network depth with a linear scaling strategy and a maximum drop rate of 0.1.

## A4   Model architectures

**UNet**.   We construct a 4-level convolutional UNet with a base channel number of 34. The total parameter count is 3.47 M.

**FNO-2d**.   For Fourier neural operator (FNO), we use 5 2-D FNO layers with a channel number of 32. The Fourier modes are set to (#$\text{Mode}_z$=32, #$\text{Mode}_x$=10). The final projection head is CONV1$\times$1(256)-GELU-CONV1$\times$1(2). The total parameter count is 3.29 M.

**F-FNO**.   For factorized Fourier neural operator (F-FNO), we use 12 F-FNO layers with a channel number of 48. The Fourier modes are set to (#$\text{Mode}_z$=70, #$\text{Mode}_x$=40). The final projection head is CONV1$\times$1(256)-GELU-CONV1$\times$1(2). The total parameter count is 3.16 M.

`NeurOLight`.   For our proposed `NeurOLight`, we use 12 F-FNO layers for tunable MMIs and 16 layers for etched MMIs with a base channel number $C$=64. The convolution stem is BSConv3$\times$3(32)-BN-ReLU-BSConv3$\times$3(64)-BN-ReLU, where BSConv is blueprint convolution [13]. The Fourier modes are set to (#$\text{Mode}_z$=70, #$\text{Mode}_x$=40). The channel expansion ratio in the FFN is set to $s$=2. The final projection head is CONV1$\times$1(256)-GELU-CONV1$\times$1(2). The total parameter count is 1.58 M.

## A5   Animation of `NeurOLight`

