# OpenReview forum: "NeurOLight: A Physics-Agnostic Neural Operator Enabling Parametric Photonic Device Simulation"
_NeurIPS.cc/2022/Conference — NeurIPS 2022 Accept_

### Official Review · Reviewer_1Stt · 2022-07-07

**Rating:** 7
**Confidence:** 4
**Soundness:** 4 excellent
**Presentation:** 4 excellent
**Contribution:** 3 good

**Summary:**

This paper proposed a physics-agnostic light field prediction framework, called NeurOLight, that consists of a joint PDE encoder and an efficient cross-shaped neural operator backbone. A superposition-based mixup technique is developed to dynamically boost the data efficiency and generalization during the training of NeurOLight. Evaluation results show that the proposed method significantly outperforms the existing methods.

**Questions:**

I am wondering if you could introduce the model architecture of the PDE encoder in more detail? Shall we know a PDE formula?

**Limitations:**

Yes, they have.

**Strengths And Weaknesses:**

Strengths:
[1] The idea is novel and interesting. It proposed a physics-agnostic light field prediction framework that can improve the efficiency and accuracy of learning parametric Maxwell PDEs.
[2] It is the first AI-based framework that can learn the terahertz light propagation inside photonic devices that generalizes to different domains.
[3] The proposed framework significantly outperforms the state of the arts with an average of 53.8% lower prediction error.

Weakness:
[1] It is not clear how much prior knowledge we should have for PDE encoder. If we do not have any prior knowledge in some applications, how does the proposed method work?
[2] It is better to report the averaged results with multiple random seeds in Table 1.

---

> ### Author Response · Authors · 2022-07-29
> **We clarify our method requires only the simplest prior knowledge and our design concepts can be applied to other problems. We added average errors over multiple trainings. Thanks a lot for your valuable feedback.**
>
> Thank you so much for your appreciation of our contributions and for providing valuable feedback on our work! Please see our responses below:
>
> **Q.1 Prior knowledge of PDE Encoder**
>
> Thanks a lot for your comments.
> * In our target problem, i.e., optical field prediction for photonic device simulation, prior knowledge about the light propagation principle is very helpful in improving the model prediction performance. This specific prior knowledge in the PDE encoder part is actually customized for our target problem, which might not be readily applicable to other tasks. However, the following design concepts and contributions are generic and can be transferred to other tasks:
>   * Our cross-shaped FNO model backbone is not limited to our problem but can be applied to other tasks for parameter-efficient PDE prediction.
>   * Insights from our PDE encoder can be applied to other applications, i.e., PDE representation is very important when training neural operators. A compact and effective representation can significantly simplify the learning process and boost generalization compared to simply feeding raw PDE parameters to the model.
>   * Data augmentation needs to be specially designed in PDE prediction tasks to mitigate the overfitting issues of pure data-driven neural operators.
> * We do not need to explicitly construct the Maxwell equation like physics-informed NNs, and we do not need to formulate complicated boundary conditions. We only need to inject the simplest and most effective prior knowledge into the PDE encoder, which is explained in Line 135-137. We find that the wave prior, light source hints, and the superposition property are the three most important prior knowledge to accomplish this field prediction task.
> * The PDE Encoder is not an NN model and thus does not have a model architecture. It simply translates the raw PDE parameters (Omega, epsilon, omega) into compact representations (epsilon, Pz, Px, H^J). The calculation of each term in the joint representation is described in each paragraph in Section 3.2.2. Our PDE encoder has a very compact and effective joint representation with negligible computational complexity.
>
> **Q.2 Average test error**
>
> Thanks for your suggestions. Due to limited computation resources, We train F-FNO[33] and our NeurOLight 4 times with different random seeds and report the average test error below. We will add the average test error and the standard deviation to the Table in the revised manuscripts. Thank you.
>
> |  Benchmark  |    Model   | Avg. test err | Std. test err |
> |:-----------:|:----------:|:------------:|:------------:|
> | Tunable MMI | F-FNO[33] |     0.302    |     0.008    |
> | Tunable MMI | **NeurOLight** |     0.129    |     0.005    |
> |  Etched MMI | F-FNO[33] |     0.521    |     0.005    |
> |  Etched MMI | **NeurOLight** |     0.389    |     0.003    |

---

> > ### Comment · Reviewer_1Stt · 2022-08-06
> > **I have read the rebuttal response**
> >
> > Thank you very much for your response and answers! I recommend this paper to be accepted.

---

> ### Author Response · Authors · 2022-08-06
> **A kind reminder of our responses to your comments. Thank you very much.**
>
> Thanks so much for your appreciation of our paper and your valuable comments. This is a kind reminder of our responses to your review comments. If you have any further questions after reading our responses, please feel free to let us know during the rebuttal window. We wish that our response has addressed your concerns. Thank you very much! We will appreciate any follow-up suggestions!

---

### Official Review · Reviewer_Fu4p · 2022-07-11

**Rating:** 7
**Confidence:** 3
**Soundness:** 3 good
**Presentation:** 4 excellent
**Contribution:** 4 excellent

**Summary:**

This paper proposes a neural operator that jointly models multiple parameters of EM simulation. Given input light wavelength, permittivity, and domain/material properties, the goal here is to establish a parametric mapping from the inputs to the output EM field via a neural network. To this end, the input domains are first normalized in terms of spatial scales and resolutions. Then, the normalized inputs are encoded as wave prior similar to the Fourier positional encoding. Following the masked image modeling allows us to cast the challenging optics simulation problem into a synthesis problem for missing regions given the input light. The network architecture for doing that is inspired by the Fourier network with a separable modification applied to that: using two 1D FFT instead of having one 2D FFT. The authors also propose an augmentation method that uses a linear combination of input lights/output fields as additional inputs/outputs.

**Questions:**

- The wave prior method reminds me of the positional encoding which has been extensively used in coordinate-baesd neural implicit functions. I wonder how these two could be related.

- The authors compare the proposed method with FDFD solution that runs on CPUs instead of GPU. Is this because the FDFD on GPUs is challenging to be adopted for this problem?  Discussion on this part would be helpful for general readers to understand the computational challenges in EM simulation.



**Limitations:**

The authors adequately addressed the limitations.

**Strengths And Weaknesses:**

Overall I like this paper. It tackles an important, but a challenging problem with novel solutions. Execution is also excellent. I hope below comments could be helpful to refine the paper.

Strength
- The paper tackles an emerging problem of learning-based optics simulation. The proposed method achieves SoTA performance on the device level simulation in terms of both accuracy and speed.
- The paper is very well written with clear descriptions and figures.
- Execution of the paper is of high quality.

Weaknesses
- There is no evaluation on the overfitting which often occurs in neural optical operators.
- Evaluation is limited on a device scale as mentioned by the authors.
- The comparison of the proposed method running on a GPU is done against FDFD method running on CPUs as described in Fig1.
- L104: It would be helpful to write detailed forms A,x, and b in the main paper

---

> ### Author Response · Authors · 2022-07-29
> **We add more discussions on the model generalization, limitation on device scales, compare with CPU simulators, and positional encoding as suggested. Thanks a lot for your valuable feedback.**
>
> Thank you so much for your appreciation of our contributions and for providing valuable feedback on our work! Please see our responses below:
>
> **Q.1 Evaluation of the overfitting**
>
> Thanks for your comments. The overfitting issue is a major challenge for data-driven neural operators. In table 1, we show both the training error and test error to show the generalization gap. Since our model is parameter efficient and adopts dropout layers during training, our proposed method shows a good generalization. Also, our proposed superposition-based mixup method can effectively augment the training distribution to improve generalization, and we evaluate its effectiveness in Table 3. Figure 11 also shows that our method can generalize to unseen MMI sizes and frequencies. Due to the unavoidable domain gap between different photonic devices, direct inference on new devices will suffer from high prediction error out-of-distribution, as we show in Figure 12. We can resolve this issue with finetuning and quickly adapt to new devices with a few training examples.
>
> **Q.2 Evaluation is limited on the device scale**
>
> Thanks for your comments. MMIs with different device scales (within a reasonable range) can be generalized by our trained model, as shown in Table 1 and Figure 11. However, a significantly larger device size will have a large data distribution shift, e.g., 5x5 MMI is much larger than 3x3 MMI. That is why we choose to apply finetuning to adapt our model to new device scales. However, if the device size is too large, we might need to increase the image size, otherwise the prediction granularity might become too coarse. Handling large devices or even challenging circuit-level simulation tasks will be our future work. We will add more discussion on this in the revised version. Thank you.
>
> **Q.3 Compare with CPU simulators**
>
> Thanks to the highly optimized PyTorch APIs, NeurOLight can benefit from >200x speedup. We compare with CPU simulators with the following considerations:
> * Leveraging the massive parallelism of GPUs is our fundamental motivation to use NN for optical field prediction.
> * We enable multi-threading for the CPU-based numerical simulator for a fair comparison.
> * Solving Maxwell numerically is equivalent to solving a large-scale sparse linear system. Large-scale sparse linear system solving is currently not supported by PyTorch/Tensorflow. It requires considerable effort to write a CUDA-based numerical solver.
>
> **Q.4 Detailed form of A, X, b**
>
> Thanks for your suggestions. This linear equation constructed for the Maxwell PDE is quite complicated [1,2]. The A matrix is a huge sparse matrix which is derived based on the PML and boundary conditions. The eigen light source vector b is different from light source J and is obtained by solving another linear equation to find the eigen mode of the input light source. The detailed form of those terms is out of the scope of this paper and can be found in the source codes of the simulator. We will try to add more introductions in the supplemental material.
>
> Reference:
>
> * [1] Hughes, Tyler W. and Minkov, et al., “Adjoint Method and Inverse Design for Nonlinear Nanophotonic Devices,” ACS Photonics, 2018.
> * [2] Advanced Computation: Computational Electromagnetics FDFD, url: https://empossible.net/wp-content/uploads/2019/08/Lecture-4f-FDFD-Extras.pdf.
>
> **Q.5 Wave prior vs. Positional encoding**
>
> This is actually a very interesting question. Previous attention-based NNs adopt linear or cosine positional encoding to inject spatial information into the attention operation. However, our wave prior is fundamentally different from a simple positional encoding.
>
> The total electromagnetic field consists of the source field and the scattering field. Directly predicting the high-frequency total field is pretty challenging for NNs. We design this wave prior to explicitly integrate the high-frequency source field into the PDE encoder to help simplify this PDE prediction task, such that our model only needs to learn how that wave scatters and interferes within the device.
>
> Our wave prior not only encodes the position information (x and z) but, most importantly, embeds the wave propagation mechanism, as we illustrated in Fig. 4. If one just directly inputs the raw permittivity distribution to the model, the model will have a hard time learning its actual physical meaning, which is harmful for generalization. Since we already know how permittivities impact light propagation, we can feed that prior knowledge to the PDE encoder to boost generalization.
>
> In Fig. 10(b), we compare various PDE encoding methods, including “positional encoding”. Comparing the first two rows, we can conclude that the pure positional information might not be useful, at least in our model. Actually, the most important thing is instead how to encode the physical meaning of permittivities into the model.
>
> We will add more discussion on this topic in the revised manuscript. Thank you!

---

> > ### Comment · Reviewer_Fu4p · 2022-08-07
> > **Discussion on the CPU-version comparison**
> >
> > I appreciate the authors' detailed explanation. One thing that I'm still in doubt about is the comparison to the CPU versions.
> > Why writing a CUDA-based numerical solver for the proposed sparse linear system is "fundamentally" challenging, especially given that NeuRIPS is a computer-science conference? Are any special numerical operators we are missing in CUDA?  The structure of the target linear system is particularly hard to be implemented in CUDA? I do understand the easiness of using Python-based frameworks such as Pytorch/Tensorflow, but it is hard to agree that this can be a reason to only compare to the CPU version. If this is the case, I think the related argument in the paper needs to be modified by toning down the speed claim.

---

> ### Author Response · Authors · 2022-08-06
> **A kind reminder of our responses to your comments. Thank you very much.**
>
> Thanks so much for your appreciation of our paper and your valuable comments. This is a kind reminder of our responses to your review comments. If you have any further questions after reading our responses, please feel free to let us know during the rebuttal window. We wish that our response has addressed your concerns. Thank you very much! We will appreciate any follow-up suggestions!

---

### Official Review · Reviewer_vy9K · 2022-07-11

**Rating:** 6
**Confidence:** 3
**Soundness:** 3 good
**Presentation:** 3 good
**Contribution:** 3 good

**Summary:**

This paper proposes NeurOLight, a variant of neural operators that is suitable to emulate optical devices efficiently. While the core behavior of the proposed NeurOLight is similar to the Fourier neural operators (FNOs), the authors introduce several techniques for optical simulations, including the scale-adaptation for merging variant domains, wave priors for Maxwell PDE encoders, masked image modeling for the light source, cross-shaped FNO that separates the horizontal and vertical pattern predictions, and superposition-based data augmentations. These all techniques for NeurOLight can streamline the complexity of the model as well as generalize the predictive performance. The authors show the proposed NeurOLight outperforms other baselines such as U-Net and FNO. They also provide some ablation studies that can support the proposed techniques are indeed helpful.

**Questions:**

The proposed method seems to be sufficiently solid and complete for its purpose. What prevents me from recommending the clear acceptance of this paper is that, as I mentioned in Strengths and Weaknesses, the main limitation of this paper is that the targeted problem is niche.

**Limitations:**

The authors do not explicitly state the limitations of their proposed method.

As I mentioned in Strengths and Weaknesses, I think that the main limitation of this paper is that the targeted problem is less relevant to the NeurIPS audiences.

**Strengths And Weaknesses:**

The paper has many merits: the introduced ad-hoc techniques are convincing for the targeted domain, the writing is clear, and the experimental validation is solid. The paper also demonstrates several interesting results including spectrum analysis and domain transfer learning. Without a doubt, it is a good application paper for optical device simulations.

My main concern is that the target domain seems to be a niche problem for NeurIPS audiences. Because the proposed techniques are generally applicable to such a niche domain, the relevance and significance of this paper are not entirely clear.

Overall, I think this paper is worthy of publication at some venues (maybe a PDE-related workshop or computational physics journals), but the venue does not have to be NeurIPS main conference.

---

> ### Author Response · Authors · 2022-07-29
> **We clarify that our framework has various machine learning contributions with important insights that fit the interest of NeurIPS general audience. Thanks a lot for your valuable feedback.**
>
> Thank you so much for your appreciation of our contributions and for providing valuable feedback on our work! Please see our responses below:
>
> **Q.1 Relevance to NeurIPS**
>
> Our proposed learning framework is an important step in ‘AI for optics’ and focuses on novel learning-based methodologies for device simulation, which is aimed at pushing forward the application of next-generation photonic computing. Though photonics is a relatively new topic to NeurIPS, we believe our work will attract the interest of the NeurIPS audience due to the following reasons:
> * Photonic deep learning has gained much momentum in recent years, which is a promising technology for next-generation efficient AI. Prior NeurIPS and other machine learning conferences also have publications in the field of photonic/physics AI [1-5]. Fast and reliable device simulation is a critical step in the photonic AI chip design process. Our ultra-fast simulation enables accelerated photonic AI chip design closure and closes the loop of optics-AI synergy.
> * This paper is self-contained, and we have a clear problem definition, i.e., we want to learn parametric frequency-domain Maxwell PDEs. We try to simplify the photonics part so that readers could understand our ML contributions even without a deep understanding of the physical mechanism or advanced optics background to understand our ML contributions.
> * Solving Maxwell PDE using learning methods is a popular topic in the ML community, which shows significant performance improvement over conventional CNNs. Our customized NeurOLight framework has intellectual contributions to the ML community and shows several insights in this direction:
>   * The importance of PDE encoding and representation in neural operator
>   * Parameter-efficient neural operator design
>   * Augmentation in the generalization of data-driven PDE learning
>   * Domain adaptation for neural operator-based PDE model
>
> We believe our work will inspire more follow-on investigations both in “AI for circuit design automation” and “efficient NN-based PDE solving”.
>
> References
>
> * [1] Julien Launay, Iacopo Poli, Kilian Muller, Igor Carron, Laurent Daudet, Florent Krzakala, Sylvain Gigan, “Hardware Beyond Backpropagation: a Photonic Co-Processor for Direct Feedback Alignment, “ NeurIPS workshop 2020.
> * [2] Jiaqi Gu, Hanqing Zhu, Chenghao Feng, Zixuan Jiang, Ray Chen, David Z. Pan, “L2ight: Enabling On-Chip Learning for Optical Neural Networks via Efficient in-situ Subspace Optimization,” NeurIPS 2021.
> * [3] Ruben Ohana, Hamlet J. Medina Ruiz et al., “Photonic Differential Privacy with Direct Feedback Alignment, “NeurIPS 2021.
> * [4] Sidharth Gupta, Remi Gribonval, Laurent Daudet, Ivan Dokmanic, “Don't take it lightly: Phasing optical random projections with unknown operators,” NeurIPS 2019.
> * [5] Jiaqi Gu, Chenghao Feng, Zheng Zhao, et al., “Efficient On-Chip Learning for Optical Neural Networks Through Power-Aware Sparse Zeroth-Order Optimization,” AAAI 2021.
>
> **Q.2 Limitations**
>
> Thanks for your great suggestions. We discussed the current method's limitations and our future direction in Line 351-353. NeurOLight is currently designed to handle several key optical devices. In our ongoing work, we will extend our framework to handle more device types as well as challenging circuit-level simulation tasks. We will add a paragraph in the revision to discuss our limitations and future directions. Thank you.

---

> > ### Comment · Reviewer_vy9K · 2022-08-08
> > **Reply to the authors**
> >
> > I appreciate the authors’ thorough response to my questions. After reading the authors' response, now I think the paper’s contribution outweighs my initial concerns (especially regarding the significance). I still think the tackled problem of this paper is not very relevant to the general ML community, but seems to be sufficiently significant for physicists and engineers who are intersted in dealing with wave PDEs by using ML. I increased my review score (and contribution score) accordingly.

---

> ### Author Response · Authors · 2022-08-06
> **A kind reminder of our responses to your comments. Thank you very much.**
>
> Thanks so much for your careful reading of our paper and your valuable comments. This is a kind reminder of our responses to your review comments. If you have any further questions after reading our responses, please feel free to let us know during the rebuttal window. We wish that our response has addressed your concerns, and turns your assessment to a more positive one. Thank you very much! We will appreciate any follow-up suggestions!

---

### Meta-Review · Area_Chair_PN7m · 2022-08-29

**Recommendation:** Accept
**Confidence:** Less certain

**Metareview:**

The authors propose a domain-specific extension of neural operators that is appropriate for photonics applications. This is an interesting application of neural operators which demonstrates the usefulness of building in physical priors. Some reviewers expressed concern about the topic being too far outside the usual focus of NeurIPS, but there is also an upside to introducing novel application areas to the NeurIPS community. All reviewers agreed the work was of high quality and worth accepting, so I recommend acceptance.

**Award:**

No

---

### Decision · Program_Chairs · 2022-09-14

Accept